# The Fight against Infection and Pain: Devil’s Claw (*Harpagophytum procumbens*) a Rich Source of Anti-Inflammatory Activity: 2011–2022

**DOI:** 10.3390/molecules27113637

**Published:** 2022-06-06

**Authors:** Nomagugu Gxaba, Madira Coutlyne Manganyi

**Affiliations:** Department of Biological and Environmental Sciences, Faculty of Natural Sciences, Walter Sisulu University, PBX1, Mthatha 5117, South Africa; ngxaba@wsu.ac.za

**Keywords:** Devil’s claw, *Harpagophytum procumbens*, clinical studies, anti-inflammatory, analgesic

## Abstract

*Harpagophytum procumbens* subsp. *procumbens* (Burch.) DC. ex Meisn. (Sesame seed Family—Pedaliaceae) is a popular medicinal plant known as Devil’s claw. It is predominantly distributed widely over southern Africa. Its impressive reputation is embedded in its traditional uses as an indigenous herbal plant for the treatment of menstrual problems, bitter tonic, inflammation febrifuge, syphilis or even loss of appetite. A number of bioactive compounds such as terpenoids, iridoid glycosides, glycosides, and acetylated phenolic compounds have been isolated. Harpagoside and harpagide, iridoid glycosides bioactive compounds have been reported in countless phytochemical studies as potential anti-inflammatory agents as well as pain relievers. In-depth studies have associated chronic inflammation with various diseases, such as Alzheimer’s disease, obesity, rheumatoid arthritis, type 2 diabetes, cancer, and cardiovascular and pulmonary diseases. In addition, 60% of chronic disorder fatalities are due to chronic inflammatory diseases worldwide. Inflammation and pain-related disorders have attracted significant attention as leading causes of global health challenges. Articles published from 2011 to the present were obtained and reviewed in-depth to determine valuable data findings as well as knowledge gaps. Various globally recognized scientific search engines/databases including Scopus, PubMed, Google Scholar, Web of Science, and ScienceDirect were utilized to collect information and deliver evidence. Based on the literature results, there was a dramatic decrease in the number of studies conducted on the anti-inflammatory and analgesic activity of Devil’s claw, thereby presenting a potential research gap. It is also evident that currently in vivo clinical studies are needed to validate the prior massive in vitro studies, therefore delivering an ideal anti-inflammatory and analgesic agent in the form of *H. procumbens* products.

## 1. Introduction

Inflammation is a multifaceted bio-physiological host defense mechanism against harmful stimuli such as unwanted pathogens, toxic substances and tissue damage [1,2,3,4,5,6]. A body of evidence has shown that inflammation has been associated with several diseases including rheumatoid arthritis, asthma, atherosclerosis and osteoarthritis (OA) [7,8,9,10,11,12,13]. Inflammation drives or modulates several disorders, which have a significant occurrence in people around the world [14,15]. Chronic diseases are ranked as some of the biggest risks to human health [16,17]. Pain associated with inflammation is a common signal in response to tissue injury [18,19]. Moreover, this has attracted significant attention as a leading cause of global health challenges [20]. Pain as an advanced stage of inflammation is an unpleasant sensation that signals successive nerve fibers and conscious sensitivity [21,22]. Currently, most nonsteroidal anti-inflammatory drugs (NSAIDs) are expensive and may lead to side-effects such as progressive heart failure, gastrointestinal lesions, and renal and liver failure [23,24,25]. In order to ameliorate the side effects, researchers have shifted towards various natural alternatives.

Inflammation and pain-related disorders possess a global health challenge. Across evolutionary adaptation time, inflammation is a physiological host defense mechanism to harmful stimuli such as tissue damage, unwanted pathogens, toxic substances and even irradiation [18]. The primary aim of inflammation is to prevent the spread of infection and to begin the recovery process. In addition, tissue homeostasis will be restored by healing and regeneration of tissues; hence returning to a normal structural and functional condition. Usually, inflammation is categorized into acute and chronic inflammation based on distinctive pathways. Numerous studies have reported that chronic inflammation is associated with several diseases, such as Alzheimer’s disease, obesity, rheumatoid arthritis, type 2 diabetes, cancer, and cardiovascular and pulmonary diseases. Currently, an overwhelming 60% of chronic disorder fatalities are a result of chronic inflammatory diseases worldwide [15]. Chronic diseases are ranked as some of the biggest risks to human health [16,17]. Pain associated with inflammation is a common signal in response to tissue injury. Inflammation and pain-related disorders have attracted significant attention as leading causes of global health challenges, hence the continued pursuance of an ideal natural anti-inflammatory agent.

For many years, ethnobotanical practices of native South Africans have become an integral part of the country’s healthcare system with an uninterrupted historic use [19]. In such a developing country, the majority of the population utilizes medicinal plants for their primary healthcare needs, thus reducing the overwhelmed, high-cost modern healthcare system. This has sparked interest in medicinal plant research since the global megatrend is towards a sustainable, eco-friendly, greener, and healthier lifestyle. One such valuable plant is *Harpagophytum procumbens* subsp. *procumbens* (Burch.) DC. ex Meisn. Historically, the native Khoisan people have harvested and utilized the Devil’s claw (*Harpagophytum procumbens*) for childbirth, loss of appetite, and as a purgative, as well as the treatment of various diseases and ailments such as menstrual problems, indigestion, bitter tonic, inflammation febrifuge and syphilis [26,27].

The *Harpagophytum* genus is widely identified as Devil’s claw belonging to the sesame seed family, Pedaliaceae which is a family of 22 genera with 90 species [28]. It is further characterized as a perennial tuberous plant with visually appealing fruits [29]. Many lengthy protrusions with sharp, grapple-like hooks grow spontaneously on the fruits and along the upper surface with two straight thorns, and so it is aptly named “Devil’s claw” [30]. However, the name Devil’s claw refers to *H. procumbens* (Burch.) DC. ex Meisn. and *H. zeyheri* Decne which are closely related species [31]. In addition, Devil’s claw (*H. procumbens*) as a traditional medicinal plant is popular for its wide range of medicinal applications. It is native to Southern Africa particularly; South Africa, Namibia, Botswana, Angola, Zambia, Zimbabwe, and Mozambique [32].

Expanding on the above, topical application of Devil’s claw ointment has been used for the treatment of boils, sprains, and sores, as well as to ease childbirth [31]. It has been used both internally and externally as a bitter tonic and for its anti-inflammatory qualities. An extensive range of secondary bioactive compounds from Devil’s claw have been identified, such as amino acids, carbohydrates, iridoids, flavonoids, and phytosterols [30,33]. Findings from drug product development reveal that terpenoids, iridoid glycosides, glycosides and acetylated phenolic extracted from tubers of Devil’s claw were utilized in the manufacture of drugs [34]. Several studies have shown Devil’s claw as an excellent source of anti-inflammatory, antibacterial, antifungal, antiviral and anticancer therapeutic properties. Scientific research also established that Devil’s claw has more advantageous benefits when compared to non-steroidal anti-inflammatory drugs (NSAIDs), in fact, it is a better alternative treatment [35,36]. It is evident from previous research findings on this plant that it was utilized as a remedy for inflammatory conditions, and it has pain-relieving properties for the treatment of degenerative painful rheumatic conditions. 

Furthermore, the herbal plants have been utilized for indigestion, blood diseases, fevers, sprains and rheumatic conditions [37]. The plant has been promoted as a food supplement for degenerative arthritis conditions [38]. Considering the number of research papers demonstrating the various biological activities of *Harpagophytum procumbens* (Devil’s claw); the next logical step should be the investigations using animal studies, as well as human trials with negative control parameters. Therefore, this paper provides a comprehensive systematic summary of current studies (published in 2011–present) conducted on the anti-inflammatory properties, bioactive compounds, and safety profiles of Devil’s claw for pain-relieving properties and for the treatment of degenerative painful rheumatic conditions as described in Section 3.6 and Section 3.7.

## 2. Methodology

The present literature review was compiled by systematically collecting, reviewing, and assembling current (2011 to present) information from globally recognized electronic databases such as Google Scholar, Scopus, Web of Science, PubMed and ScienceDirect. This comprehensive search was conducted using keywords (Devil’s claw; *Harpagophytum procumbens*; Clinical Studies; Biological activities; Safety; in vivo). In addition, we pre-screened the abstracts before studying the full documents. The literature review was analysed in-depth to acquire new insights and knowledge gaps for prospective research opportunities. The current review covers only one subspecies particularly subsp. *procumbens* and throughout the review *Harpagophytum procumbens* subsp. *procumbens* (Burch.) was referred to as *Harpagophytum procumbens*.

## 3. Results

Various search engines were utilized to extrapolate 996 articles from Google Scholar, 53 from Scopus, 14 from PubMed, 7 from Web of Science and 47 from Science Direct. Among these, only articles published in English between 2011 and 2022 were selected according to the inclusion/exclusion parameters in Figure 1. A preliminary screening of titles and abstracts was conducted to omit any irrelevant studies, followed by a full text review and analysis. This body of research provides significant information from the main findings over the past ten years and explores the changes, gaps, and future prospects. 

### 3.1. Botanical Description of Harpagophytum procumbens

*Harpagophytum procumbens* (Devil’s claw) is one of the medicinally and economically important members of the sesame seed family, Pedaliaceae [39]. It is a weedy, perennial, tuberous plant with a conspicuous fruit [40]. The fruit is the one that gave this plant its colloquial name ‘Devil’s claw’ [26,41]. The fruit is a woody capsule of varying sizes with long protruding sharp, grapple-like hooks [30]. Inside the fruit are numerous seeds in rows of four in each loculus and these are dispersed over a long period of time. It has creeping, annual stems that grow to a height of 2 m from a tuberous fleshy rootstock [41]. This stem has hollow branches which are covered with glandular hairs that exude slimy, sticky sap. This stem sprouts from a succulent taproot [11]. 

The creeping nature of the stem is what gave this plant its species name *procumbens*. From the succulent tap root, thick secondary roots branch off horizontally. These secondary tubers are 25 cm long with 6 cm thickness. The particular secondary roots are the therapeutic part of the plants as they consist of between 0.5 and 3% of iridoid glycosides [30,42]. The leaves are borne on opposite sides of the trailing stem, and they are irregularly divided into 3–5 lobes while being greyish-green in colour because of the tiny whitish mucilage cells covering them as shown in Figure 2A. The flowers are tubular shaped with a color ranging from dark velvety red or purple to pink a simple, opposite, and oval in shape while the tube base and mouth are yellowish to white. They open during the day to allow pollination by bees [41]. The genus *Harpagophytum* consists of two related species *procumbens* and *zeyheri* [28,29]. Of the two species, *H. procumbens* is considered to be more medicinally efficient than *H. zeyheri*. The superiority of *H. procumbens* over *H. zeyheri* was attributed to the presence of higher chemical constituents in *H. procumbens* [30]. This has led to medicinal product(s) derived from it being sold as one or the other or mixed [28].

It was then proposed that the two species be turned apart as this mix-up was found to lower the quality of the intended medical drug [30]. The two species were then distinguished based on fruit capsule seed rows. In *H. procumbens,* the fruit capsule has four seed rows whereas that of *H. zeyheri* has two [40]. This way of distinguishing the two species of *Harpagophytum* was not effective because the seed numbers were not restricted to four or two as there might have been introgression between the two species [41]. The other distinguishing feature of these species was the appearance of the fruit capsule. The fruit capsule in *H. procumbens* usually has three long arms while that of *H. zeyheri* has eight short arms with slightly curved arms at the tips [41]. However, this distinction was also not sufficient, particularly at harvest time. Once the tubers are harvested, the distinction between the tubers was virtually impossible. Mncwangi et al. [43] compared the chemical composition of the two species and found that they were not comparable. This chemical composition still confirmed that *H. procumbens* was still superior to *H. zeyheri*. The name *Harpagophytum* came from the Greek word ‘harpago’ which means ‘grappling hook’ [44,45]. There are several common names that have been given to this plant these include Devil’s claw, grapple plant, wood (Teufffelskralle), Trampelklette (German), Griffe du diable (French) (Kempel).

**Figure 2 molecules-27-03637-f002:**
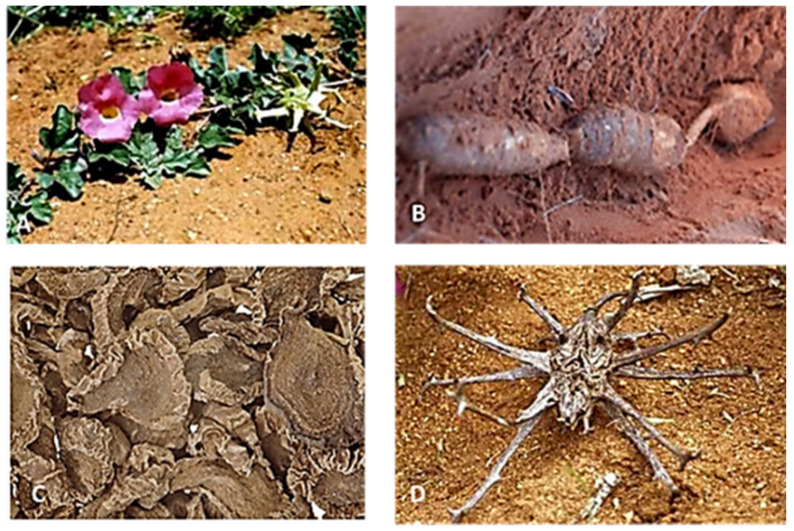
*Harpagophytum procumbens* (Devil’s claw) (**A**) Devil’s claw plant with pink flowers (SANBI) [44]; (**B**) *H. procumbens* tubers; (**C**) Dried *H. procumbens* sliced tubers used for plant extracts [27] (**D**) Devil’s claw (SANBI) [45].

### 3.2. Traditional Uses of Harpagophytum procumbens as Bases of Investigation

For centuries, the first inhabitants of Southern Africa, also well known as the indigenous “Khoisan” people, utilized Devil’s claw in ancient prescriptions for various health conditions [11,26]. The high influx of immigration from other African tribes including Bantu-speaking people brought about the exchange of traditional knowledge systems [46]. The arrival of Europeans in South Africa was no exception, Devil’s claw tubers were traded to Germany in 1962 [28]. As the years progressed, various herbal formulations such as powders, infusions, tinctures, extracts, and decoctions have been prepared from *Harpagophytum procumbens* root tubers [47]. *Harpagophytum procumbens* remedies are an excellent source of topical medicines. For these uses, the tubers were collected in the wild, sliced, dried, and made into tea [48,49]. They have also been used as a topical treatment for burnt wounds, sprains, sores, boils, and other skin problems [30]. As a topical treatment, the root is dried and powdered and directly used to cover wounds; or the powdered root is mixed with animal fat or petroleum jelly to make an ointment that effectively treats muscular aches and pains, and painful joints [11]. However, there has not been definitive scientific evidence to substantiate the effective use of the Devil’s claw as an ointment [31].

It was regarded to be a therapeutic pain reliever as well as treating arthritis. In addition, Devils claw has been utilized for treating a large selection of conditions, including urinary tract infections, sores, fever, dyspepsia, blood diseases, postpartum pain, sprains, and ulcers [50]. Due to its anti-inflammatory and analgesic effects, the use of *H. procumbens* has spread to other countries, United Kingdom, Holland, the United States and the Far East where it is registered as a food supplement for degenerative arthritic conditions, in Germany and France it is an important herbal medicine [26]. In Brazil, *H. procumbens* is released as herbal medicine (by the National Surveillance Agency, ANVISA) used for the relief of mild joint pain and acute back pain [32]. *Harpagophytum procumbens* in these health conditions is used in the formulations of an infusion, decoction, tincture, powder, and extract [30]. The utilization of *Harpagophytum procumbens* has the function of a mild laxative, small quantities alleviate menstrual cramps although higher quantities aid in expelling retained placentas. One of the earliest ethnobotany records was Samuel Kariko using Devil’s claw to combat constipation, cough, diarrhea, and venereal infections.

The secondary tuber of *Harpagophytum procumbens* is dried and pulverized into powdered form and used externally for dressing wounds and sometimes infused with animal fat or Vaseline^®®^ to produce a wound- and burn-healing ointment [11,51]. *Harpagophytum procumbens* is also used to treat digestive problems, diabetes, as a tonic herb and infectious ailments [52,53]. A liquid decoction of the *H. procumbens* secondary tubers or chewing it might result in stomach and postpartum pain relief, as previously reported by the Topnaar people of Namibia [54]. Furthermore, *H. procumbens* infusions have therapeutic effects for ‘blood diseases’ and as a bitter tonic [55]. With emerging ailments and/or diseases on the rise, *H. procumbens* has been reported to alleviate asthma, pancreatitis, tuberculosis, and for treating liver diseases, syphilis, rheumatism, kidney diseases and gonorrhea [11]. In a cleansing ceremony, an ancient practice, the Devil’s claw secondary tubers are mixed with Asparagus or Protasparagus roots and used to cover the cuts on the skin [51]. The ethnobotanical accounts gathered have been from the earliest records and solidify the basis for an *H. procumbens* investigation.

### 3.3. Phytochemistry of Harpagophytum procumbens

The phytochemical studies of *H. procumbens* have revealed that the main chemical constituents that have been thought to reduce inflammation in this herb are iridoid glycosides (IG), phenolic glycosides such as acteoside, isoacteoside and bioside, harpagoquinones, amino acids, phenolic (aromatic) acids (caffeic, cinnamic and chlorogenic aicds), flavonoids, phytosterols and carbohydrates (turanose and starchyose) as bioactive compounds of this plants [30,33,38]. Iridoid glycosides are a large group of compounds belonging to a class monoterpene derived from geraniol with a general form of cyclopentopyran although, in certain instances, the ring is cleaved and forms secologanin [56]. Iridoid glycosides are a massive group of compounds in *H. procumbens* with high concentrations of harpagosideh, arpagide, procumbide, procombiside 8-O-(p-Coumaroyl)-harpagide, and harpagogenine as shown Figure 3 [30,57]. These compounds were isolated in tuber roots and are believed to exhibit anti-inflammatory as well as analgesic properties [58]. The first iridoid glycoside to be isolated in 1962 by a scientist from Wϋrzburg was harpagoside. Among the iridoid glycosides, harpagoside is the most investigated and is considered a reference standard for titration purposes [38]. It is believed that a single chemical constituent cannot carry out the bioactivity, hence, the contribution of several active compounds is required. This might explain the findings that harpagoside was not an efficient anti-inflammatory agent. The chemical structures of these iridoid glycosides were elucidated using nuclear magnetic resonance (NMR) spectroscopy [59]. Studies have shown that *H. procumbens* root tuber extracts that inhibited inflammation and alleviated pain contained harpagide [60], harpagoside, 8-p-coumaroylharpagide and acteoside as the main bioactive compounds [30,34]. From these results, it was concluded that the efficacy of the extract is dependent on the presence of the active agents. Interestingly, an extract free of harpagoside also exhibited marked inhibition of inducible NO synthase expression, however, pure harpagoside exerted strong antioxidant activity [61,62].

Phenolic glycoside refers to any molecule with a sugar unit attached to a phenol aglycone in a general sense [63].

Phenolic glycosides have been indicated in the therapy of inflammatory diseases [64]. The phenolic glycosides that have been identified in *H. procumbens* are 8- feruloyharpagide, verbascoside, leuxosceptoside, pagoside, illustrated in Figure 4A–C [65]. Among these, verbascoside, also known as acteoside, was found to have a strong anti-inflammatory effect which was either comparable or even higher than that of harpagoside [66]. The other phenolic glycosides of *H. procumbens* either have a strong antioxidant activity or antimicrobial activities [43]. Orally administered amino acids such as glycine, lysine, L- proline and valine have exhibited anti-inflammatory activities in some tests [67]. Amino acids found in plants have been reported to exhibit anti-inflammatory activity [68]. The minor chemical constituents of *H. procumbens* such as flavonoids (kaempferol and luteolin) and phenolic or aromatic acids (such as cinnamic, caffeic and chloronic acids, Figure 4A,B) have also been reported to exert some anti-inflammatory abilities [43]. Flavonoids were isolated using phytochemical screening and confirmed with 1D NMR spectra in these plants [69]. Terpenoids, ursolic, oleanolic acid and beta-sitosterol were the other minor chemical constituents of *H. procumbens* that have been found to have anti-inflammatory effects. The isolation of these chemical constituents from other plant parts has been considered laborious and can hinder pharmacological studies [47].

### 3.4. Anti-Inflammatory Properties of Harpagophytum procumbens

The *Harpagophytum procumbens* ethanol extract (60% *v*/*v* ethanol) reduced the COX-2 mRNA quantity in a dose-dependent manner at 50 μg/mL and 200 μg/mL(Figure 5). The *Harpagophytum* extract was Pascoe^®®^-Agil 240 mg, film-coated tablets. The transcription factor activator protein 1 (AP-1) activity was reduced in murine macrophages and cytokine expression. Similarly, TNF-α and interleukin 6 (IL-6) were suppressed at concentrations of 100 μg/mL and 200 μg/mL, respectively. The results also showed the most anti-inflammatory potential of *Harpagophytum procumbens* extract (IC_50_) at 100 mg/mL and lower [39]. Harpagoside and harpagide, which are monoterpenoids commonly found in *Harpagophytum procumbens*, have been shown to inhibit TNFα-secretion in PMA-differentiated THP-1 cells. Subsequently, mRNA expression belonging to a particular protein in leukocyte transmigration was induced. After 48 h, the expression was retained on high concentrations and strong induction of L-selectin and PSGL-1 of stimulation. Within 3 h, there was a positive impact on TNFα and ICAM-1 mRNA expression. These findings indicate that cell movement into the swollen tissue is due to the harpagoside and harpagide showing their immune-modulatory role and resident macrophages were elevated indicating a potentially effective anti-inflammatory agent [70].

Hostanska et al. [71] also demonstrated the anti-inflammatory activity of Devil’s claw tuber extracts by an external metabolic activation and the discharge of proinflammatory cytokines. The results showed no significant metabolic activation of the extract mixture S9 mixture (S9 mix homogenate blended with 60% ethanol *H. procumbens* extract) with no cytotoxic and inhibition activities. However, TNF-α demonstrated potential anti-inflammatory activity with EC_50_ concentration levels of 116 ± 8.2 μg/mL and 49 ± 3.5 μg/mL for DC and DCM (*p* < 0.01), respectively. TNF-α, IL-6 and IL-8 dose-dependently were inhibited in monocytic THP-1 cells treated with LPS at non-cytotoxic doses (50–250 μg/mL). The cytokines’ effect was not influenced by metabolic activation, interestingly, even though we also found that the concentration of harpagoside and caffeic acid derivates was reduced. The anti-inflammatory potential of harpagoside and harpagide was studied by structural molecular biology and computer-assisted drug design, molecular docking. However, evidence suggests that harpagoside and harpagide affect COX-2, with binding energies estimated at −9.13 and −5.53 kcal/mol, respectively. The stabilization of harpagoside and harpagide occurred at the active site of COX-2 through 7 and 10 hydrogen bonds, respectively. It can be concluded that harpagoside and harpagide are promising leads for additional study as potential anti-inflammatory/analgesic compounds; highly selective COX-2 inhibitors. They might provide a safer more effective anti-inflammatory/analgesic agency than the currently commercialized non-steroidal anti-inflammatory drugs [72].

An ex vivo rat colon model has demonstrated the anti-inflammatory activity of *H. procumbens* root tuber extracts in multiple tissues and ultimately provides a therapeutic alternative to inflammatory diseases [54]. In another study, Gyurkovska and colleagues [67], reported that several *H. procumbens* root extracts exhibited excellent anti-inflammatory activities in vitro systems. Murine macrophages displayed the release of nitric oxide (NO) and cytokine (TNF-α, IL-6) and the expression of COX-1 and COX-2 [67]. There is now compelling evidence that harpagoside, a natural compound from Devil’s claw, has been shown to possess anti-inflammatory activities. In a mouse model, harpagoside suppressed inflammation-induced bone loss while preventing the receptor activator of nuclear factor κB ligand (RANKL)-induced osteoclastogenesis. Furthermore, harpagoside prevented the development of osteoclasts from mouse bone marrow macrophages dose-dependently. In the same content, it reduced extracellular signal-regulated kinase (ERK) and c-jun N-terminal kinase (JNK) phosphorylation, resulting in the prevention of Syk-Btk-PLCγ2-Ca2+ in RANKL-dependent early signaling [73]. In addition, harpagoside also demonstrated its ability to suppress c-FOS functioning as AP-1 transcription factors in osteoarthritis chondrocytes [62].

In another study, *H. procumbens* ethanol extracts from the whole plant, as well as other medicinal plants, were subjected to anti-denaturation action in heat-treated Bovine Serum Albumin (BSA), as a function of anti-inflammatory compounds. The dose-dependent results showed the inhibition of protein (albumin) denaturation at concentrations of 50–1000 µg/mL. *H. procumbens* demonstrated good anti-inflammatory properties [74]. In a study conducted by Cock and Bromley [75], *H. procumbens* root tuber extracts were shown to be effective bacterial trigger inhibitors of autoimmune inflammatory diseases [75]. In a most recent study, a randomized triple-blind placebo-control trial was conducted on runners with self-reported knee pain using *H. procumbens* combined with other medicinal plants as an analgesic and anti-inflammatory agent. The database on the study showed a significant reduction in leg fat mass (*p* = 0.037) and knee thermograms noticeable in IG (*p* < 0.05). No differences were found in the overall safety and the efficacy of *H. procumbens* mixtures as an anti-inflammatory nutritional supplement [76]. Mariano et al. [38] investigated the antiarthritic properties of *H. procumbens* root extract and molecular mechanisms, as well as bioactive compounds. Interestingly, the expression of the Cannabinoid receptor 2 (CB2) receptor was enhanced by certain bioactive compounds, however, it was unexpressed in the osteoarthritic tissues. Chronic inflammation has been recognized as a variable mediator of osteoarthritis. Several studies have reported that CB 2 receptors are expressed as an immune response and display an inflammatory effect [77,78,79,80].

### 3.5. Analgesic Effects of Harpagophytum procumbens

Current synthetic drugs used for the management of analgesia have been expensive as well as ineffective. These drugs have been associated with numerous adverse effects, hence the endless pursuance of substitute sources. For centuries, native tribes across the world have used medicinal plants for therapeutic pain relief, and nowadays this has gained attraction in the research as well as pharmaceutical sectors [3,13,81,82]. Parenti et al. [21], administered several combinations of *H. procumbens* extract with morphine via in vivo trials using rats. The rats were induced with chronic constriction injury (CCI) as neuropathic pain. The data suggested a synergistic result of the combination of *H. procumbens* and morphine. Furthermore, the combination acts as an excellent pain reliever, as well as reduces the side effects related to neuropathic pain [21]. In a recent study, a cream formulation was prepared from *H. procumbens* root extract for the treatment of neck/shoulder sport-related pains. The treatment was administrated on the skin of healthy participants. After 2 weeks, the pain subsided and the participants reported a boost in strength, mobility and working abilities [83].

In a Cochrane review [84], daily quantities of *H. procumbens* with 50 mg or 100 mg harpagoside administrated to participants improved the pain. While other reports showed the effectiveness of lesser doses of *H. procumbens* in a smaller trial group [84]. Previously, male SD rats were subjected to plantar incision and spared nerve injury (SNI) to determine the pain-related behavior and *H. procumbens* was utilized as a treatment. There was a significant increase in the MWT values of the 300 mg/kg *H. procumbens*-treated group followed by a decline in the number of 22–27 kHz USVs after 6 h and 24 h of plantar incision operation. The 300 mg/kg dose treatment of *H. procumbens* extracts prepared from the whole plant displayed continuous improvement of SNI-induced hypersensitivity responses by MWT after 21 days. The study recommends the use of *H. procumbens* extracts as a potent analgesic agent for the treatment of acute postoperative pain and chronic neuropathic pain [85]. In a traditional healing system, herbal medicine usually consists of more than one medicinal plant, thus the concoction of formulations. Radomska-Leśniewska et al. [85] supported this hypothesis by preparing Reumaherb concoctions from *Harpagophytum procumbens* mixed with two other herbal plants. The study found that oral administration of 1.2 mg Reumaherb to mice exhibited potential anti-inflammatory, as well as anti-angiogenic, effects on mononuclear cells. It was, however, concluded the Reumaherb concoctions showed excellent concentrations of angiogenesis which might be used for rheumatoid arthritis patients [86].

### 3.6. Diseases Associated with Inflammation and Pain

Rheumatoid arthritis, by definition, is a chronic inflammatory disorder that might affect the lining of joints and induce painful swelling in humans, which can lead to bone degradation and joint deformity [87,88]. The illness can harm a range of body systems in some people, including the skin, eyes, lungs, heart, and blood vessels. Osteoarthritis (OA), on the other hand, is defined as a type of articular disease that results from the degeneration of articular cartilage and subchondral bone [61,89]. It is a common condition in elderly people [90]. OA is often manifested by joint pain, tenderness, and limitation of movements. Rheumatoid arthritis and osteoarthritis inflammation are managed using pharmacological and non-pharmacological approaches; the traditional approach targets the treatment of symptoms connected to diseases such as pain and physical dysfunction, whereas non-steroidal anti-inflammatory drugs (NSAIDs) have anti-inflammatory and analgesic effects [21]. Nonetheless, these drugs are very expensive, and they carry substantial side effects such as gastrointestinal, cardiovascular, liver and renal complications [85]. These adverse effects have forced patients to look for alternative treatments that are effective and safe, such as herbal and nutritive supplements. *H. procumbens* was introduced as Teltonal, a herbal analgesic that does not only relieve and treat chronic inflammation [21]. In vitro studies have revealed that *H. procumbens* root tuber extracts reduced rheumatoid and osteoarthritis inflammation by suppressing pro-inflammatory mediators such as interleukin (IL)-induced production of metalloproteinase [30]. The effectiveness of Devil’s claw tuber extracts in reducing pain and inflammation caused by rheumatoid arthritis and osteoarthritis has been attributed to the presence of active ingredients [91].

*H. procumbens* has also been used in the treatment of ulcerative colitis, one of the chronic disorders of the digestive tract [54]. The lining of the colon becomes inflamed, resulting in small open sores or ulcers that generate pus or mucus. Patients suffering from ulcerative colitis show symptoms that vary from mild to widespread inflammation. These include abdominal discomfort or cramps, rectal bleeding, diarrhoea, anaemia, fatigue, fever, nausea, weight loss, loss of appetite, abdominal sounds, mouth ulcer, loss of body fluids and nutrients, skin lesions and growth failure in children [92]. The interest in using extracts prepared from *H. procumbens* root tuber was triggered when the administering of medicines such as 5-amino salicylate, azathioprine, 6-mercaptopurine, cyclosporine, sulfasalazine, and antitumor necrosis factor (TNF)-α inhibitors failed to yield desirable results [93]. Treatment with these medicines was accompanied by several side effects with some of them being severe.

Extracts from *H. procumbens* were prepared with traditional or biocompatible solvents (water and hydroalcoholic solution) to offer efficacious and safe treatments. The extracts reduced inflammation by inhibiting pro-inflammatory mediators such as oxidative stress, 5-HT, PGE and 8-iso PGF2α. These effects were found to be comparable with sulfasalazine (2 µg/mL) which was used as a reference drug. The antioxidant and anti-inflammatory effects of *H. procumbens* extracts were ascribed to the following secondary metabolites; harpagoside, gallic acid, catechin, epicatechin, resveratrol, phenolic and flavonoids. This extract was also reported to act on bacterial and fungal strains that are often associated with colon inflammation such as *Candida albicans* and *C. tropicalis*. The presence of *C. albicans* and *C. tropicalis* in the colon has been reported to aggravate colitis’ clinical symptoms [94]. Moreover, a proteomic analysis of the colon that has been treated with *H. procumbens* extract revealed that there was an up-regulation of the levels of peroxiredoxin-2(PRDX2), glutathione reductase, (GSHR) catalase (CAT), and superoxide dismutase (SOD) which are known to counteract oxidative stress-induced organ injury [93]. Studies have shown that the antioxidant activity of *H. procumbens* contributes to its anti-inflammatory effect [95]. It has been established that *H. procumbens* extracts also exert anti-inflammatory effects on renal inflammation [26]. Renal inflammation is one of the common problems associated with kidney diseases that have been affecting people these days [96]. Some clinical trial studies have shown the efficacy of *H. procumbens* in chronic diseases for human treatment [97], and other studies are summarized in Table 1.

### 3.7. Safety Aspects of Harpagophytum procumbens

Medicinal plants have been utilized to heal a variety of infectious and non-infectious ailments and/or diseases since ancient times, despite the fact they also exude toxic effects. The toxicity of medicinal plants is affected by several factors such as the exposure time, climate, the potency of bioactive metabolites, various plant parts, genetic variations within the species, dosage consumed, soil, and individual body chemistry [16,105]. The literature search has shown the dramatic decline in *Harpagophytum procumbens* studies and its safety investigations are no exception. Prior to 2011, there were many studies conducted on the acute and chronic toxicity of *Harpagophytum procumbens* plant extracts and their bioactive compounds. Murine peritoneal macrophages and 3-(4,5-dimethyl-thiazol-2-yl)-2,5-diphenyl tetrazolium bromide (MTT) were used to investigate the cytotoxicity of *H. procumbens* extracts and their purified substances. Compound harpagide demonstrated the highest toxic result at 1 mg/mL concentration by decreasing the number of viable cells as compared with β-OH-verbascoside and martynoside [66]. In a recent study, Joshi and colleagues [26] conducted a toxicological study of *H. procumbens* aqueous-alcohol root tuber extract using a rat model. In this model, various doses of *H. procumbens* extracts were administrated to male and female Sprague Dawley rats for a few months (one and three months). Histopathology results showed an increased occurrence of heart inflammation in the control groups as compared to treatment groups. Despite this, there were no detected histopathology results due to the intake of the extracts [26].

In another study, acute and chronic toxicity were investigated to detect the effect of *H. procumbens* (Devil’s claw) capsules on male and female mice. Some adverse effects such as the growth of forelimb inflammation and snout alopecia were experienced by one male animal, and no significant mortality was recorded. The results showed a mild decline in locomotive effects of mice served with 1 and 3 g/kg concentration of *H. procumbens.* The blood glucose concentrations were lower in the treated mice than in the control group. The overview of this study suggests lower doses of Devil’s claw should be administered since it will result in lower toxicity [106]. Furthermore, the consumption of medicinal plants might induce nephrotoxicity which results in a rapid decline in kidney function, for instance, the effect of *H. procumbens* on renal function [107]. Bano et al. [108], reported no mortality and morbidity in mice after oral administration of Devil’s claw to determine the acute toxicity. In another study, the pharmacokinetic parameters of harpagoside were assessed using single-dose, two-treatment on six horses in a randomized cross-over design.

Moreover, *H. procumbens* plant extracts were regarded as unsafe in a study harpagoside the main compound of Devil’s claw, was administrated to horses at 5 mg/kg and 10 mg/kg concentrations. However, no clinical side effects were demonstrated [109].

Davari [110], established that a 600 mg/kg concentration of *H. procumbens* extract causes necrosis in the kidney, liver, and lungs of fetuses of pregnant mice. Nonetheless, there were no changes in defects in body weight, structural malformations, and crown-rump length of treated embryos. The findings show the teratogenic potential, as well as the histopathological alterations in fetal tissues [110]. Several clinical studies on the safety as well as efficacy effects of *H. procumbens* were conducted in animals as well as humans (Table 2 and Table 3). It is worth noting that Menghini and colleagues [11] have provided a mountain of evidence supporting the therapeutic efficacy of *H. procumbens* preparation in both preclinical and clinical studies.

## 4. Conclusions

*Harpagophytum procumbens* is a highly valued medicinal plant in southern Africa as well as the entire world. The body of evidence has shown that Devil’s claw plants are richly endowed with anti-cholesterolemic, antioxidant, anti-inflammatory, and pain killing effects for the therapeutic treatment of countless diseases including rheumatoid arthritis, memory loss, lower back pain, osteoarthritis, lumbago, Syphilis, diabetes, indigestion, and heartburn, as well as acting as a detoxifying and tonic agent. Toxicological data have shown that *H. procumbens* has adverse effects in high concentrations and various models promote low dose levels as a safety parameter. Despite the massive body of evidence of in vivo studies on the anti-inflammatory and analgesic activity of *H. procumbens,* alternative lines for future studies focus on animal and human models. Due to its popularity, Devil’s claw has been well-documented since the 1800 s, nonetheless, the research data on the anti-inflammatory and analgesic activity of *H. procumbens* has reduced since the 2000 s. However, this review has identified the decline in research data and promotes further investigation regarding the above. In order to streamline the future application of *H. procumbens,* more clinical studies are required to close the gap in research. Hence, providing increased therapeutic efficacy and safer *H. procumbens* products in the health and pharmaceutical sectors.

## Figures and Tables

**Figure 1 molecules-27-03637-f001:**
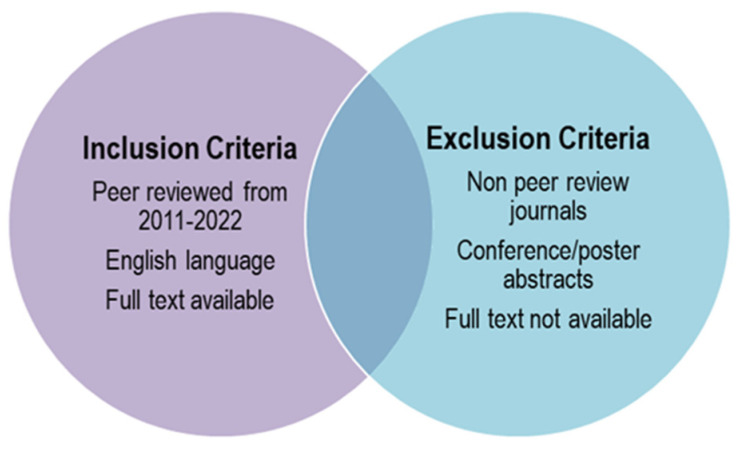
Inclusion and exclusion criteria on the systematic literature search.

**Figure 3 molecules-27-03637-f003:**
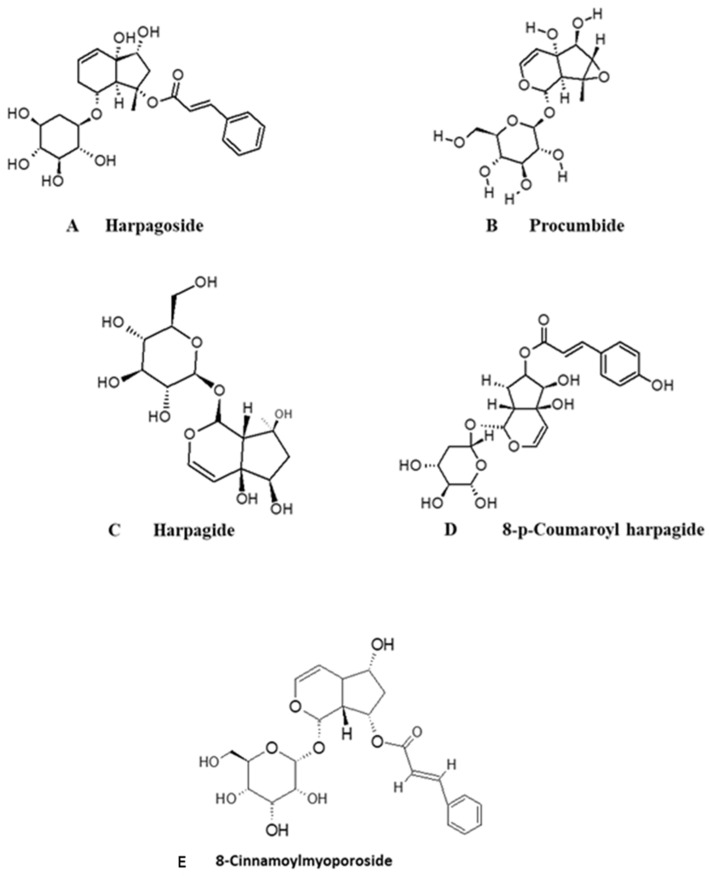
Chemical structures of the major iridoids glycosides and phenylethanoid from *H. procumbens* [45].

**Figure 4 molecules-27-03637-f004:**
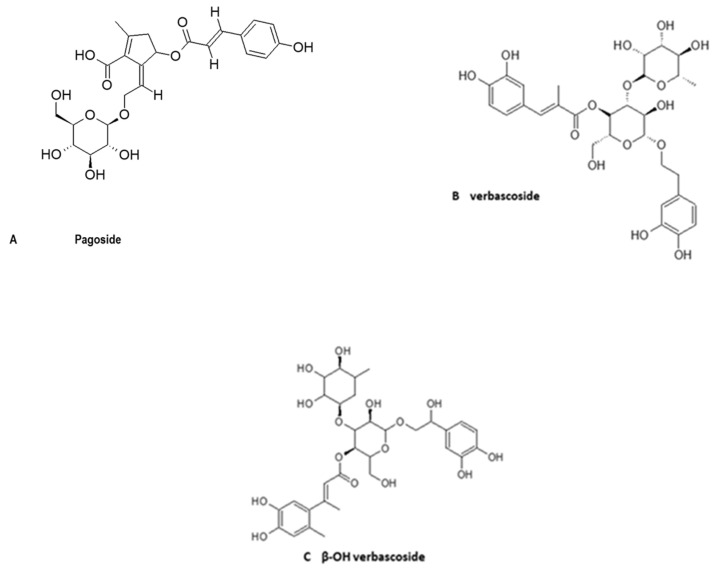
Chemical structure of the phenolics; phenolic glycosides of *H. procumbens* [29].

**Figure 5 molecules-27-03637-f005:**
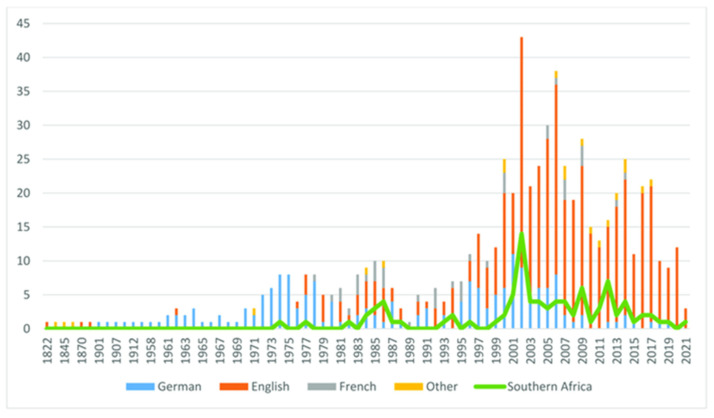
Number of research papers on *Harpagophytum* spp., 1822–2021 (publication language/origin of research) [28].

**Table 1 molecules-27-03637-t001:** *Harpagophytum procumbens* as an anti-inflammation and pain reliever associated with various diseases.

Name of the Disease	Disease Symptoms or Complications	Part(s) Used	Biological Effects of Devil’s Claw	Compound Constitutes	Reference
Alzheimer’s disease	Loss of memory and cognitive judgment	Dried roots	Anti-Alzheimer effect	Verbascoside derivatives	[98]
		Dried roots	Management of the clinical symptoms related to ad, inflamed tissues	Harpagoside	[99]
		Roots	Anti-inflammatory activity	Phenylethanoid glycoside	[62]
Rheumatoid arthritis	Pain associated with joints, back, or muscles. Joints are tension, swelling, tenderness, or weakness	Harpagoside compound	Anti-inflammatory activity	Harpagoside	[61]
		Roots	Antiarthritic effects, anti-inflammatory activity	Harpagoside	[38]
		Harpagoside compound	Anti-inflammatory activity	Harpagoside	[61]
		Harpagoside compound	Antirheumatic effects, pain reliever, anti-inflammatory activity	Harpagoside,	[100]
		Roots	Anti-inflammatory activity	Harpagoside	[101]
Osteoporosis	Lowered bone pain, height loss and muscle spasms.	Plant material	Anti-osteoporotic activity	Harpagide	[102]
		Plant material	Anti-osteoporotic activity	Harpagoside	[103]
		Harpagoside compound	Anti-inflammatory	Harpagoside	[73]
		Roots	Anti-inflammatory, antioxidant, analgesic	Unidentified	[20]
		Roots	Pain reduction and function	Unidentified	[90]
Diabetes	Injury to the large blood vessels of the heart, brain and legs	Roots	Anti-inflammatory and antirheumatic properties	Harpagoside	[19]
Obesity	Overweight, exhaustion, pot belly, or breathe heavily	Roots	Suppress appetite	Iridoid glycosides	[104]
Psoriasis	Hives, dryness, flakiness, peeling, redness	Roots	Anti-inflammatory	Phenylethanoid glycosides verbascoside	[51]

**Table 2 molecules-27-03637-t002:** Pre-clinical studies of Harpagophytum procumbens.

Disease/Study Aim	Drug Formulation	Participants/Animals	Outcomes	Reference
Neuropathic pain	Plant extracts	Rats	Reduce pain	[21]
Toxicity	Plant extracts	Rats	Significant sex-related effects on blood chemistry	[26]
Toxicity	Devil’s claw capsules	20 mice	Decrease in blood glucose level, weight gain in female	[106]
neurotoxicity	Capsules	Rats	anti-inflammatory activity and antioxidant effects	[111]

**Table 3 molecules-27-03637-t003:** Clinical studies of Harpagophytum procumbens.

Disease/Study Aim	Drug Formulation	Participants/Animals	Outcomes	Reference
Knee Osteoarthritis	Tablets	Sixty human patients	No significant difference but consider to be safer.	[97]
Gonarthritis	Root extracts	Ninety-two human patients	Reduced gonarthritis symptoms	[112]

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
