# Peer review of "The Fight against Infection and Pain: Devil’s Claw (Harpagophytum procumbens) a Rich Source of Anti-Inflammatory Activity: 2011–2022"

_molecules, 2022, doi:10.3390/molecules27113637_

Round 1
Reviewer 1 Report
The manuscript entitled "The fight against infection and pain: The Claw of the Devil (Harpagophytum procumbens): Rich source of anti-inflammatory activity: 2011-2022" provides an in-depth review on the activities of title plant againt inflammation.
This is well written manuscript however needs major revision before acceptance.
- The genus, species and subspecies names should be in italics. They are properly written in some places but represented in not-italicized fonts in many sections such as Lines 61, 56, 11, etc.
- How amany subspecies are there for H. procumbens? Is this review on a whole species of P. procumbens covering all subspecies (as written in title) or a review of single subspecies (subsp. procumbens) as written i Line 11. It shoule be clarly mentioned and corrected.
- Line 95, authors mention section 4 but section 4 is concluding remark. What is its relevance?
- Line 42 and other places, please check all references are years carefully.
- Line 47, are all NSAIDs expensive? It should not be generalized.
- Is the common name Devil's claw or Claw of the Devil? They mean same thing but it should one standard in thsi paper. The title and other sections use them differently.
- Figure 3, B cinnamic acid's structure is not correct.
- Caffec acid and cinnamic acid are not phenolic glycosides. They are only phenolic compounds.
- Fig. 4, Strcuture of keampferol is not correct. Double bonds in B ring are missing.
- Fig. 4, B. is this Glycine?
Reviewer 2 Report
Several review articles were published about this important medicinal plant. Even though your article is skillfully written, it lacks some important citations.
1) Some important previous review articles. Please see the link below, as an example. After you do so, the uniqueness of you article is highlighted.
https://pubmed.ncbi.nlm.nih.gov/17128436/
2) Some important research articles, such as H.R. Farpour et al., that you cited one of their works.
https://www.hindawi.com/journals/ecam/2021/5596892/
3) My humble opinion is that Figure 4 is unnecessary since these are VERY well known compounds that can be found in almost all plants.
Reviewer 3 Report
The review ”The fight against infection and pain: The Claw of the Devil (Harpagophytum procumbens): Rich source of anti-inflammatory activity: 2011-2022 “ is very interesting and reports the innumerable biological properties of Harpagophytum procumbens. To complete the excellent work done, I ask the authors to write a paragraph on any “in vivo” antitumor activities of the plant and possibly of the new molecules found in it.
Round 2
Reviewer 1 Report
Manuscript has been revised as suggested in few sections and became better but may sissues are remaining as below:
1) I had suggested authors to check and correct structures but not to delete as they are main compounds. Just deleting them wound not improve mansucript as there are still mistakes in structures.
2) Structures in figure 2A and 2C are not correct as the sugar moiety attched is not glucose. Current structure contains xylose. They should be checked throughly and corrected.
3) Structure of 3A pagoside is also not correct.
4) Structures of 3B and 3C need revision. Please check carefully.
5) Figure 3C is also an iridoid having phenolic acidf moeity. It should be shifted to figure 2.
6) Major issue: Line 434-456, authors mention about clinical trials but most of the studies included in Table 1 are invitro ones. If there are clinical studies, they should be represented in separate section and with proper table and description.
Author Response
|
Reviewers Comment |
Authors Feedback |
Page no and Line |
|
1) I had suggested authors to check and correct structures but not to delete as they are main compounds. Just deleting them wound not improve mansucript as there are still mistakes in structures. |
I have corrected and replaced the structures back in the document. |
Whole document |
|
2) Structures in figure 2A and 2C are not correct as the sugar moiety attched is not glucose. Current structure contains xylose. They should be checked throughly and corrected. |
I have corrected Figure 2A and 2C using PubChem |
Pg 6, line 235 |
|
3) Structure of 3A pagoside is also not correct. |
3A pagoside was corrected |
Pg 6, line 235 |
|
4) Structures of 3B and 3C need revision. Please check carefully. |
3B and 3C was corrected |
Pg 7, line 244 |
|
5) Figure 3C is also an iridoid having phenolic acidf moeity. It should be shifted to figure 2. |
Figure 3C was moved |
Pg 6, line 235 |
|
6) Major issue: Line 434-456, authors mention about clinical trials but most of the studies included in Table 1 are invitro ones. If there are clinical studies, they should be represented in separate section and with proper table and description. |
Table 2 was added and sentence “Several clinical studies with safety effects of H. procumbens were conducted in animals as well as human (Table 2).” |
Pg 13, line 493 |
Round 3
Reviewer 1 Report
I have provided my sincere comments for revision of this manuscript many times already but I think authors are not taking these comments seriously or do not want to revise.
My comments specially regarding plant name itself and chemical structures are never answered properly.
For example, although authors have changed plant name to Devil’s claw in manuscript text, the title still includes Claw of the Devil. Authors should use correct name.
Regarding chemical structures, it looks like there is confusion about basic structures of iridoids and phenolic glycosides.
As I had already commented many times, Fig 3. C b-OH verbascoside is not an iridoid. It is a phenylethyl glycoside. It is a derivative of verbascoside which is included in Figure 4.
However, the structure of this structure is still incorrect.
Line 256-258, the explanation about phenolic glycosides is not correct. As name suggest they are glycosides of phenolic compounds. It does not need to have two aromatic groups and two sugars.
I had already commented that the structure of pagoside is not correct. Authors have commented that they revised but it is sill same. In current structure what authors have provided, the sugar moiety is not glucose.
Structure of 4D verbascoside is also not correct.
Compound 4E 8-cinnamoylmyoporoside is an iridoid derivative. I had already commented that.
Section 3.6 already covers analgesic activity. It is again repeated in 3.7.
The title of 3.7 is also not correct.
Anti-inflammatory effect is already explained in section 3.5. Why is it again repeated in 3.7?
Table 1 has many mistakes. Are these disease symptoms (complications) or side effects?
What is relation of anti-inflammatory and antirheumatic activity in diabetes?
Table 2 is mixed up. Why are authors mixing pharmacokinetic activity and toxicity study?
Table 3. Hypertension: The reference 44 “Cuspidi, C.; Sala, C.; Tadic, M.; Grassi, G.; Mancia, G. Systemic hypertension induced by Harpagophytum procumbens (devil's claw): a case report. J. Clin. Hypertens. 2015, 17, 908–910. “ suggests that this plant induces hypertension and it is not a study about antihypertensive activity.
Reference 112 “Devil's claw (Harpagophytum procumbens) ameliorates the neurobehavioral changes and neurotoxicity in female rats exposed to arsenic “ is conducted in female rats. How can it be considered as a clinical study?
